# Defects in the GINS complex increase the instability of repetitive sequences *via* a recombination-dependent mechanism

**Malgorzata Jedrychowska, Milena Denkiewicz-Kruk, Malgorzata Alabrudzinska, Adrianna Skoneczna⏺, Piotr Jonczyk, Michal Dmowski⏺\*, Iwona J. Fijalkowska⏺\***

Institute of Biochemistry and Biophysics, Polish Academy of Sciences, Warsaw, Poland

\* mdmowski@ibb.waw.pl (MD); iwonaf@ibb.waw.pl (IJF)

**Data Availability Statement:** All relevant data are within the manuscript and its Supporting Information files.

## Abstract

Faithful replication and repair of DNA lesions ensure genome maintenance. During replication in eukaryotic cells, DNA is unwound by the CMG helicase complex, which is composed of three major components: the Cdc45 protein, Mcm2-7, and the GINS complex. The CMG in complex with DNA polymerase epsilon (CMG-E) participates in the establishment and progression of the replisome. Impaired functioning of the CMG-E was shown to induce genomic instability and promote the development of various diseases. Therefore, CMG-E components play important roles as caretakers of the genome. In *Saccharomyces cerevisiae*, the GINS complex is composed of the Psf1, Psf2, Psf3, and Sld5 essential subunits. The Psf1-1 mutant form fails to interact with Psf3, resulting in impaired replisome assembly and chromosome replication. Here, we show increased instability of repeat tracts (mononucleotide, dinucleotide, trinucleotide and longer) in yeast *psf1-1* mutants. To identify the mechanisms underlying this effect, we analyzed repeated sequence instability using derivatives of *psf1-1* strains lacking genes involved in translesion synthesis, recombination, or mismatch repair. Among these derivatives, deletion of *RAD52*, *RAD51*, *MMS2*, *POL32*, or *PIF1* significantly decreased DNA repeat instability. These results, together with the observed increased amounts of single-stranded DNA regions and Rfa1 foci suggest that recombinational mechanisms make important contributions to repeat tract instability in *psf1-1* cells. We propose that defective functioning of the CMG-E complex in *psf1-1* cells impairs the progression of DNA replication what increases the contribution of repair mechanisms such as template switch and break-induced replication. These processes require sequence homology search which in case of a repeated DNA tract may result in misalignment leading to its expansion or contraction.

## Author summary

Processes that ensure genome stability are crucial for all organisms to avoid mutations and decrease the risk of diseases. The coordinated activity of mechanisms underlying the maintenance of high-fidelity DNA duplication and repair is critical to deal with the

**Funding:** This work was supported by the National Science Centre, Poland (www.ncn.gov.pl) grant no. 2017/26/M/NZ3/01044 to MD and grant no. TEAM/2011-8/1 from the Foundation for Polish Science (www.fnp.org.pl), co financed from European Union - Regional Development Fund „New players involved in the maintenance of genomic stability" to IJF. The funders had no role in study design, data collection and analysis, decision to publish, or preparation of the manuscript.

**Competing interests:** The authors have declared that no competing interests exist.

malfunction of replication forks or DNA damage. Repeated sequences in DNA are particularly prone to instability; these sequences undergo expansions or contractions, leading in humans to various neurological, neurodegenerative, and neuromuscular disorders. A mutant form of one of the noncatalytic subunits of active DNA helicase complex impairs DNA replication. Here, we show that this form also significantly increases the instability of mononucleotide, dinucleotide, trinucleotide and longer repeat tracts. Our results suggest that in cells that harbor a mutated variant of the helicase complex, continuation of DNA replication is facilitated by recombination processes, and this mechanism can be highly mutagenic during repair synthesis through repetitive regions, especially regions that form secondary structures. Our results indicate that proper functioning of the DNA helicase complex is crucial for maintenance of the stability of repeated DNA sequences, especially in the context of recently described disorders in which mutations or deregulation of the human homologs of genes encoding DNA helicase subunits were observed.

## Introduction

Mechanisms by which organisms efficiently and faithfully control DNA stability are subjects of primary scientific interest. Mutagenesis produces genetic variations that drive the evolution of all species but at the same time may affect the lives of individual organisms, resulting in enhanced risk of carcinogenesis and other disorders [1–3]. The instability of repeated DNA sequences, also called satellite sequences, causes more than 30 disorders. Microsatellites and minisatellites are DNA motifs consisting of 1–9 or 10–100 base pairs, respectively, that are repeated from five times up to hundreds of times [4,5]. Such sequences are frequently found in genomes and are characterized by high variability. Dinucleotide repeats are the most abundant DNA repeats (48–67%) identified in many species [6,7], but in primates, mononucleotide repeats were identified as the most numerous class of simple DNA repeats [4,8]. DNA repeats influence chromatin organization, gene activity, and regulation of DNA metabolic processes. Alleles of genes carrying altered minisatellites have been correlated with a number of severe diseases, such as progressive myoclonus epilepsy [9], insulin-dependent diabetes mellitus [10], attention-deficit hyperactivity disorder [11], asthma [12], ulcerative colitis [13] and several cancer subtypes [14–16]. Expansions in trinucleotide repeats in humans can cause Huntington's disease, myotonic dystrophy, spinocerebellar ataxia, and many other neurodegenerative disorders [17–19].

Mutation rates in DNA repeats are very high ($10^{-2}$–$10^{-6}$ events per locus per generation) compared with the rates of point mutations at average gene loci ($10^{-9}$–$10^{-10}$) [2,20]. Molecular mechanisms of DNA repeat instability have been studied in many experimental systems, including bacteria, yeast, fruit flies, mice, and human cells [21]. Various mechanisms were shown to be involved in DNA repeat instability, i.e., formation of unusual DNA structures during DNA replication or slipped-strand mispairing [22–24], DNA recombination [25–27], DNA repair [28–34], and transcription [35,36]. Moreover, these mechanisms may interact with each other [37–41]. For example, DNA regions that are processed by DNA repair mechanisms and contain repeat tracts are subject to expansion/contraction by slip-strand mispairing errors upon strand invasion and formation of secondary structures during repair synthesis [42]. Other examples of factors increasing the instability of repetitive sequences are mutations in the yeast DNA polymerases or their decreased levels [43–48], mutations in genes encoding the Rad27 nuclease involved in Okazaki fragments processing, DNA ligase I, the PCNA polymerase processivity clamp or the Rfc1 subunit of the clamp loader [30,49].

In all organisms, DNA replication is carried out by the replisome, a multiprotein complex [50]. In eukaryotic cells, the highly efficient unwinding of double-stranded DNA is catalyzed by a helicase known as the CMG (Cdc45-Mcm2-7-GINS) complex. The CMG complex is a macromolecular assembly of 11 essential replication factors: the Cdc45 protein, four subunits of the GINS complex (Psf1, Psf2, Psf3, and Sld5) and the heterohexameric Mcm2-7 complex, which functions as the helicase motor. This complex translocates along the leading strand to separate the duplex DNA [51]. CMG participates in the establishment and progression of the replisome serving not only as a helicase but also as a platform for coordination of the activity of different components of the replisome [52–55]. Moreover, CMG has been shown to interact with the Pol ε complex, forming the CMG-E complex [56,57] and stimulating polymerase activity [58]. Experiments in fission yeast suggest not only structural but also functional interplay between the CMG helicase and Pol ε [59]. Recently, it was shown that functional impairment of the CMG-E complex induces genomic instability and promotes the development of genetic diseases [60].

Among many interactions within the CMG-E complex, it was shown that the Psf1 subunit of the GINS complex interacts not only with Psf3 and Sld5 but also with the essential subunit of Pol ε - Dpb2 [61,62]. It was also demonstrated that Cdc45 interacts with the GINS complex via the Psf1 and Psf2 subunits [63,64]. The *psf1-1* mutant, isolated in Hiroyuki Araki's laboratory, is temperature-sensitive and does not grow at 37°C. The Psf1-1 mutant subunit (R84G) exhibits an unchanged interaction with Dpb2, but the interaction of this subunit with Psf3 is severely impaired [54,62], which compromises the integrity of the GINS complex. However, there are no data regarding the mechanism by which this mutant form of Psf1 influences interactions with other components of the replisome and influences the physiology of the cell. We previously showed that in a *psf1-1* strain, destabilization of the GINS complex influences the level of mutagenesis, increasing the levels of both base substitutions and insertions/deletions (indels). As observed with many other replisome defects, a significant fraction of spontaneous mutagenesis in this mutant is due to increased participation of the error-prone DNA polymerase zeta (Pol ζ) [62].

In this work, we analyzed the effects of impaired interactions within GINS on the instability of DNA repeat tracts (mononucleotide, dinucleotide, trinucleotide, and others). All the DNA repetition types analyzed were much more unstable in the *psf1-1* mutant than in the wild-type strain. Our results demonstrate that in the *psf1-1* mutant, Pol ζ does not contribute to the instability of the tested repeat tracts; instead, homologous recombination (HR) is the major mechanism responsible for the observed instability. We also show increased formation of single-stranded DNA regions in the *psf1-1* cells, which is likely the result of impaired replication. To deal with this problem, recombination-associated repair synthesis may proceed, increasing the risk of instability of repetitive regions, especially those forming secondary structures. These findings highlight the importance of proper interactions within the GINS complex for the stability of repetitive sequences.

## Results

As the replisome components are highly conserved from yeasts to humans in terms of structure, chemistry, and functionality, we used *Saccharomyces cerevisiae* as a model organism to investigate the influence of the defective functioning of the essential GINS complex on the stability of DNA repeat tracts. In our experiments, we used the Psf1-1 mutant form of a subunit of this complex, which exhibits impaired interaction with the Psf3 subunit. We examined the effects of the *psf1-1* allele on the stability of a variety of repeat tracts by employing two widely

used assays, namely, the chromosomal trinucleotide repeat expansion assay [65] and the plasmid-based frameshift assay [66,67].

## The *psf1-1* allele causes increased instability of DNA repeat tracts

Trinucleotide repeats are a class of microsatellite sequences, expansions of which are responsible for more than 30 human developmental and neurological disorders [18,19,68]. Previously, it was shown that impaired functioning of the CMG-E complex induces genomic instability and promotes the development of various diseases [60,69,70], demonstrating that CMG-E components have important roles as caretakers of the genome [71]. Therefore, we sought to investigate the impact of the defective functioning of the GINS complex on the stability of trinucleotide repeats using the chromosomal assay developed in the laboratory of R. Lahue [65]. In this assay, the promoter of the *URA3* reporter gene contains 25 repeats of tested trinucleotides located between the TATA box and the transcription start site. These repeats do not interfere with the expression of *URA3* and therefore yield sensitivity to the toxic drug 5-fluoroorotic acid (5-FOA). Expansion of at least five trinucleotides (>29 repeats) results in the production of a long transcript encompassing an out-of-frame ATG and, as a consequence, in translational incompetence which can be selected on medium containing 5-FOA which is toxic to cells producing Ura3 protein [72]. Appropriate sequences containing trinucleotide repeats (CTG, GAA or TTC) or the control sequence (75 random nucleotides) in the promoter of *URA3* were introduced into the *psf1-1* mutant strain and control wild-type cells. Wild-type cells with the "scrambled" (C, A, T, G) control sequence yield 5-FOA$^R$ colonies at a rate of <0.54 x $10^{-8}$, while the *psf1-1* mutation results in increased rates of forward mutagenesis in *URA3* (19 x $10^{-8}$) (Fig 1). Wild-type cells containing (GAA)$_{25}$, (TTC)$_{25}$ and (CTG)$_{25}$ tracts yield 5-FOA$^R$ colonies at a rate of <0.44 x $10^{-8}$, <0.47 x $10^{-8}$ and 37 x $10^{-8}$, respectively, which confirms that (CTG)$_{25}$ repeats are more unstable than (GAA)$_{25}$ and (TTC)$_{25}$ tracts (Fig 1) [73]. When (GAA)$_{25}$, (TTC)$_{25}$ and (CTG)$_{25}$ expansions were measured in the *psf1-1* mutant, the mutation rates were two orders of magnitude higher than those observed for the wild-type strain (114 x $10^{-8}$, 132 x $10^{-8}$, and 5795 x $10^{-8,}$ respectively) (Fig 1). Frequent expansions of tested trinucleotide tracts in the *psf1-1* strain highlight the importance of proper interactions within the GINS complex for the stability of repeated trinucleotides.

To further investigate the level and mechanisms of DNA tract instability in the *psf1-1* strains, we employed the widely used and well-documented plasmid system developed in T. Petes' laboratory based on the reporter gene *URA3* with in-frame insertions of various mini- or microsatellite sequences [66,67]. The stability of the repeat tracts depends on their length and nucleotide composition. Therefore, we tested (G)$_{18}$ (pMD28), (GT)$_{49}$ (p99GT), (AACGC AATGCG)$_4$ (pMD41) and (CAACGCAATGCGTTGGATCT)$_3$ (pEAS20) tracts in the *psf1-1* strain and, as a control, in *PSF1* cells. As an additional control, we designed the pKK2 plasmid carrying a random sequence that was also inserted in-frame and devoid of any repeats (see Materials and Methods). The alterations within the tracts leading to out-of-frame insertions or deletions can be selected on medium containing 5-FOA [72]. All the repeat tracts, i.e., (G)$_{18}$, (GT)$_{49}$, (AACGCAATGCG)$_4$ and (CAACGCAATGCGTTGGATCT)$_3$, were roughly two- to four-fold less stable in the *psf1-1* strain than in the wild-type strain (Fig 2).

To confirm that the 5-FOA resistance of wild-type and *psf1-1* cells with the tested plasmids resulted from altered repeat tracts, we characterized independent isolates derived from each strain. Using capillary electrophoresis, we analyzed the lengths of the PCR products encompassing the repeat tracts from 5-FOA-resistant mutants (Table 1).

As expected, in both the wild-type and *psf1-1* mutant strains harboring the control pKK2 plasmid, we did not observe changes in the length of the random sequence. In these mutants,

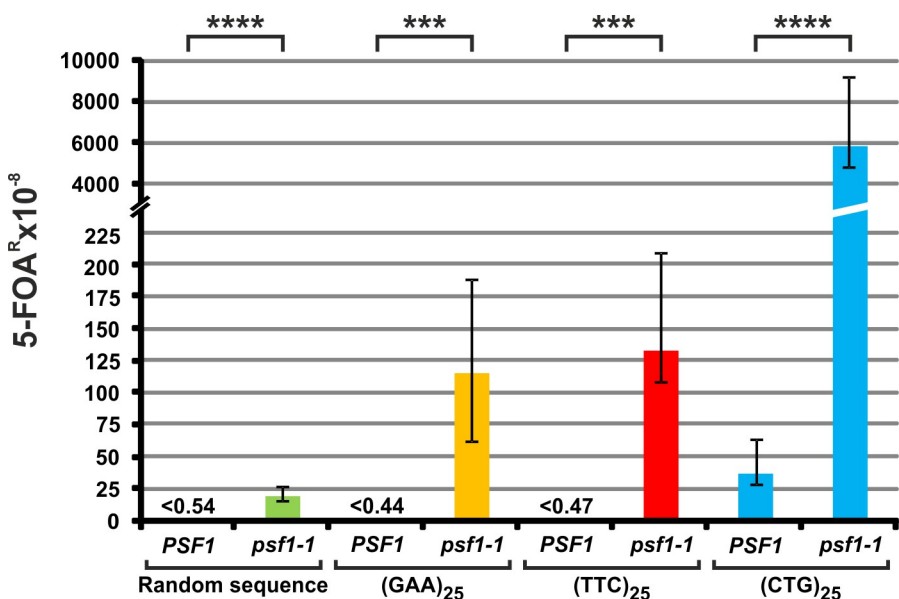

**Fig 1. The *psf1-1* allele enhances chromosomal trinucleotide repeat instability in yeast.** The 5-FOA$^R$ mutation rates were determined at 23°C for yeast cells carrying the analyzed sequences; the values are medians with 95% confidence intervals calculated from data for at least ten cultures of each strain; the *p*-values for *psf1-1* mutants versus wild-type strains were calculated using the nonparametric Mann-Whitney U statistical test (**** $p \leq 0.0001$; *** $p \leq 0.001$). All associated *p*-values are presented in the S1 Table.

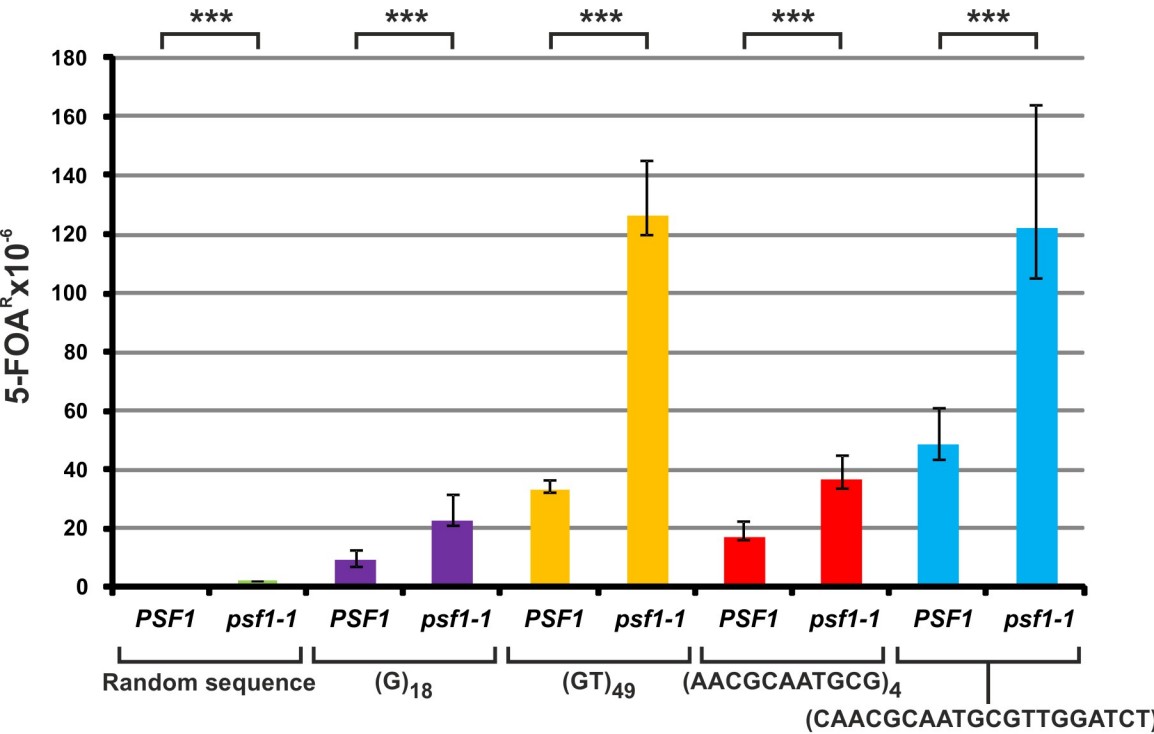

**Fig 2. The *psf1-1* allele increases the instability of various repeat tracts.** The 5-FOA$^R$ mutation rates were determined at 23°C for yeast cells with the indicated genotypes carrying plasmids with analyzed repetitive sequences; the values are medians with 95% confidence intervals calculated from data for at least ten cultures of each strain; the *p*-values for *psf1-1* mutants versus wild-type strains were calculated using the nonparametric Mann-Whitney U statistical test (*** $p \leq 0.001$). All associated *p*-values are presented in the S1 Table.

**Table 1. Types of alterations within different repeat tracts in the *psf1-1* mutant and wild-type strains carrying plasmids with repeat sequences.**

| Tract sequence | Relevant genotype | Classes of tract alterations [%][a] | | | | | | | Number of analyzed clones |
|---|---|---|---|---|---|---|---|---|---|
| | | Large deletions | -2 | -1 | 0 | +1 | +2 | Large additions | |
| random sequence | *PSF1* | | | | 100 | | | | 66 |
| | *psf1-1* | | | | 100 | | | | 96 |
| (GT)$_{49}$ | *PSF1* | 28 | | 2 | | 66 | | 4 | 85 |
| | *psf1-1* | 47 | | 1 | 1 | 44 | | 7 | 75 |
| (AACGCAATGCG)$_4$ | *PSF1* | | 81 | 8 | 9 | | 2 | | 52 |
| | *psf1-1* | | 85 | 2 | 11 | | 2 | | 47 |
| (CAACGCAATGCGTTGGATCT)$_3$ | *PSF1* | | 46 | 46 | | 8 | | | 63 |
| | *psf1-1* | | 11 | 81 | 6 | 2 | | | 54 |

a. The numbers in the column headings are the number of repeat units added (+) or deleted (-).

we observed only changes in the *URA3* coding sequence, confirming that pKK2 is an appropriate control for our experiments (Table 1). Capillary analysis indicated that the alterations in the lengths of the poly(GT) tracts mainly involved insertions of one repeat in both the wild-type (66%) and *psf1-1* (44%) strains. The second important class of alterations was the deletion of more than two repeats in both the wild-type (28%) and *psf1-1* (47%) strains. In the plasmids containing (AACGCAATGCG)$_4$ repeats, the most common alterations were two-tract deletions (81% in the wild-type strain and 85% in the *psf1-1* strain). For minisatellites with long repeat units (CAACGCAATGCGTTGGATCT)$_3$, the most frequent changes were those involving one or two-repeat deletion (92% for both strains). These types of alterations are consistent with previously published results from T. Petes' laboratory [66,67] and the predicted structures that can be formed on the mother or daughter strand to produce contractions or expansions, respectively (S1 Fig). Alterations in mononucleotide microsatellites cannot be analyzed using capillary electrophoresis because these alterations are mainly deletions or additions of one repeat, which are beyond the sensitivity range of this method. In summary, the *psf1-1* allele causes an increase in repeat tract instability compared to that seen in the wild-type strain. To further analyze the mechanism underlying repeat tract instability in the *psf1-1* strains, we decided to use a plasmid-based system, which provides the opportunity to analyze the stability of various repeat tracts.

## MMR repairs DNA loops which are at the origin of microsatellite sequences instability in *psf1-1* cells

What mechanisms are responsible for the instability observed in the *psf1-1* strains? The most common mechanism that explains tract instability is DNA/polymerase slippage [24,27,74]. Changes in microsatellite sequences resulting from rearrangement of the template and primer can be recognized and repaired by the mismatch repair (MMR) system [75]. In yeast, there are two MMR complexes composed of Msh2, Msh3, and Msh6, which are homologs of the prokaryotic MutS protein [76,77]. The Msh2-Msh6 and Msh2-Msh3 complexes are also called MutSα and MutSβ, respectively. Inactivation of Msh6 decreases the stability of repeat tracts with repeat units of 1 or 2 bp, while inactivation of Msh2 or Msh3 decreases the stability of repeat tracts with repeat units ranging in length from 1 to 14 bp [67,78,79]. None of the complexes are capable of efficiently repairing 20-base loops [67,80,81]. Mutations in genes affecting MMR dramatically reduce microsatellite stability [82–87].

To determine whether the MMR mechanism corrects the alterations in repeated sequences in the *psf1-1* mutant, we analyzed the rate of mutagenesis in derivatives carrying the *psf1-1*

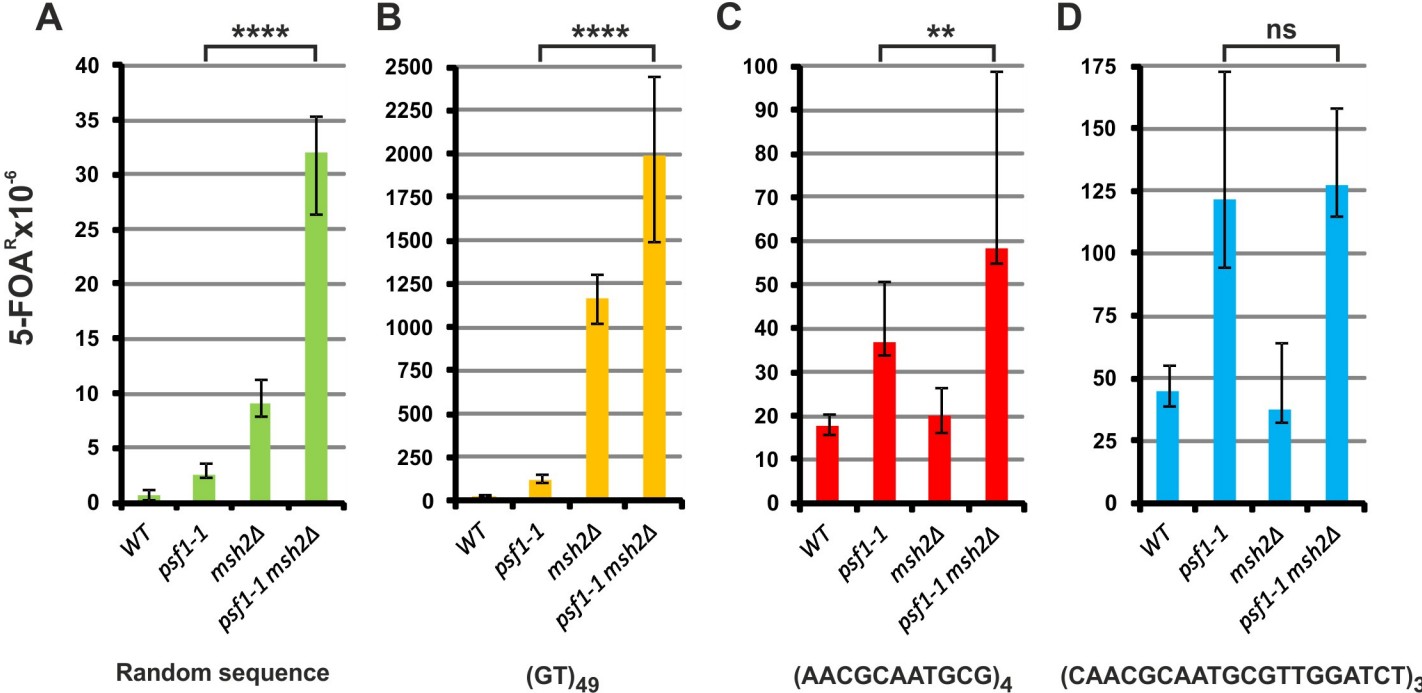

**Fig 3. Effect of mismatch repair on repeat tract stability in *psf1-1* cells (A-D).** The 5-FOA$^R$ mutation rates were determined at 23˚C for yeast cells with the indicated genotypes carrying plasmids with the analyzed sequences; the values are medians with 95% confidence intervals calculated from data for at least ten cultures of each strain; the *p*-values for *psf1-1 msh2Δ* versus *psf1-1* mutant strains were calculated using the nonparametric Mann-Whitney U statistical test (**** $p \leq 0.0001$; ** $p \leq 0.01$; ns–$p > 0.5$). All associated *p*-values are presented in the S1 Table.

allele and deletion of the *MSH2* gene (Fig 3). The *psf1-1* and *msh2Δ* single mutants carrying the random sequence on the control plasmid pKK2 show 4-fold and 15-fold mutator effects on the *URA3* coding sequence, respectively. Synergy in the mutator effect (54-fold increase) is observed in the double mutant *psf1-1 msh2Δ*/pKK2, which indicates that *psf1-1* significantly increases the number of replication errors recognized and repaired by the MMR system (Fig 3A). The *psf1-1* strain carrying a plasmid with a (GT)$_{49}$ tract shows a 4-fold elevated rate of microsatellite instability compared to that in the wild-type strain. The *msh2Δ* strain harboring the same plasmid elevates the level of microsatellite instability by 38-fold. The double mutant *psf1-1 msh2Δ* shows a 66-fold increase in instability. This destabilizing effect in the *psf1-1 msh2Δ* double mutant is stronger than an additive effect would have been (42-fold), suggesting a synergistic interaction of the two mutations (Fig 3B). Absence of the function of the MMR system has a moderate effect on (AACGCAATGCG)$_4$ tracts and no effect on relatively long tracts (CAACGCAATGCGTTGGATCT)$_3$ (Fig 3C and 3D), as expected from the specificity of the MMR mechanism. In the wild-type background, deletion of *MSH2* does not significantly affect the stability of these repeat tracts (Fig 3C and 3D). Together, these results indicate that Msh2 has an impact on microsatellite repair but is not involved in the repair of the tested minisatellite tracts in both wild-type and *psf1-1* strains.

### The elevated rate of repeat tract instability in the *psf1-1* strain is not dependent on the error-prone DNA polymerase ζ

We previously showed that approximately 50% of spontaneous mutagenesis in *psf1-1* cells is associated with the participation of Pol ζ in DNA synthesis [62]. Pol ζ is composed of four subunits, namely, Rev3 (the catalytic subunit), Rev7, Pol31, and Pol32 [88,89]. Therefore, to test

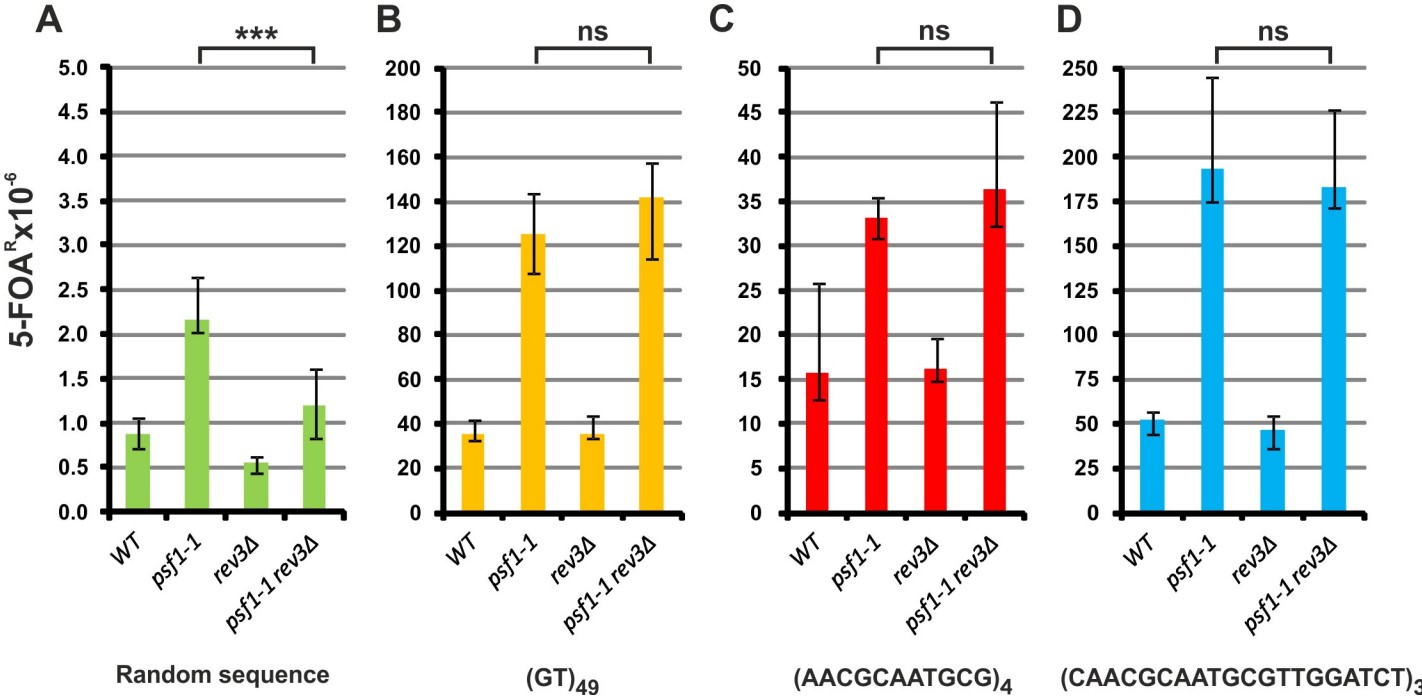

**Fig 4. Pol ζ has no effect on repeat tract stability in *psf1-1* (A-D).** The 5-FOA[R] mutation rates were determined at 23°C for yeast cells with the indicated genotypes carrying plasmids with the analyzed sequences; the values are medians with 95% confidence intervals calculated from data for at least ten cultures of each strain; the *p* values for *psf1-1 rev3Δ* versus *psf1-1* mutant strains were calculated using the nonparametric Mann-Whitney U statistical test (*** *p*≤0.001; ns–*p*>0.5). All associated *p*-values are presented in the S1 Table.

whether Pol ζ is also involved in the instability of repeated sequences in the *psf1-1* strain, we performed measurements in the *rev3Δ* background. The level of forward *URA3* mutations in strains carrying a plasmid with the random sequence decreased by 45% in the *psf1-1 rev3Δ* strain (Fig 4A). This suggests that destabilized interactions within the GINS complex in the *psf1-1* mutant lead to increased participation of Pol ζ in DNA synthesis. In both *PSF1* and *psf1-1* strains possessing plasmids with repeat tracts, we observed no effect of Pol ζ deficiency on the rates of appearance of 5-FOA[R] colonies (Fig 4B, 4C and 4D). This result suggests that Pol ζ does not participate in the mechanisms underlying the destabilization of repetitive sequences in both wild-type and *psf1-1* cells.

## HR is engaged in repeat tract instability in the *psf1-1* strain

Many DNA repair mechanisms are based on HR. This is a highly conserved mechanism for the repair of DNA double-strand breaks (DSBs) and recovery of stalled or collapsed replication forks [90–92]. However, although recombination is thought to be error free, this mechanism is also potentially mutagenic, e.g., recombination can promote the instability of repeated DNA sequences [27] or cause genome rearrangements in the cells under constant replication stress [93].

The recombinases Rad52 and Rad51 are two major players in HR [94,95]. The most important function of Rad52 is the promotion of Rad51 filament formation on RPA-coated ssDNA. To investigate the role of Rad52 in the instability of repetitive sequences in the *psf1-1* strain, we combined *rad52Δ* with the *psf1-1* allele. Both alleles exhibit a mutator effect on the overall mutation rate at *URA3*. Strain *psf1-1* with the random control sequence shows a 3.8-fold

increase in *URA3* mutagenesis, while the *rad52Δ* allele exhibits a 9.6-fold mutator effect (Fig 5A). In the double mutant (*psf1-1 rad52Δ*), forward *URA3* mutagenesis increased 14.2-fold showing an additive effect (Fig 5A).

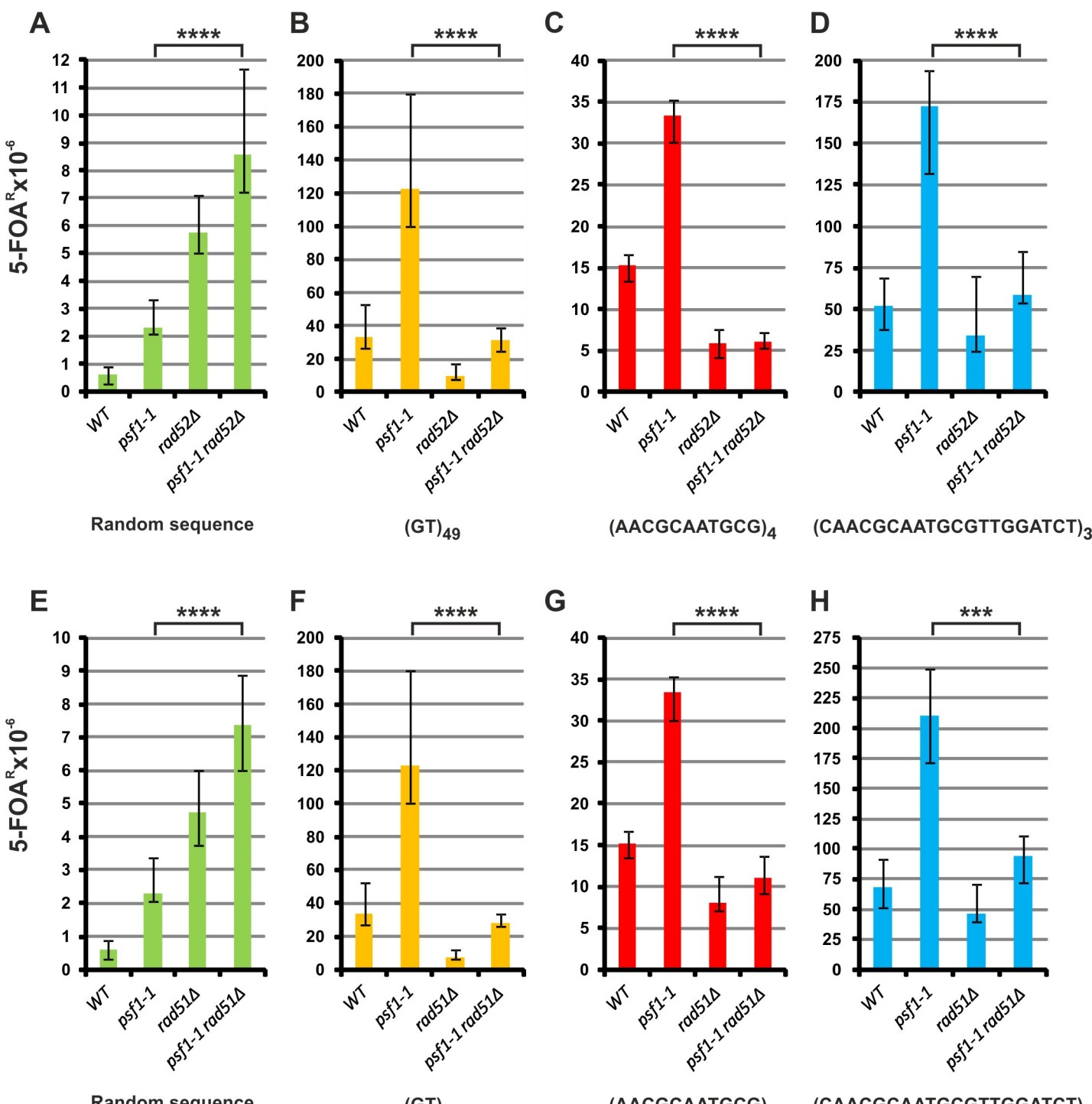

**Fig 5. The instability of repeat tracts in *psf1-1* cells depends on the recombinases Rad52 (A-D) and Rad51 (E-H).** The 5-FOA$^R$ mutation rates were determined at 23˚C for yeast cells with the indicated genotypes carrying plasmids with the analyzed sequences; the values are medians with 95% confidence intervals calculated from data for at least ten cultures of each strain; the *p* values for *psf1-1 rad52Δ* or *psf1-1 rad51Δ* versus *psf1-1* mutant strains were calculated using the nonparametric Mann-Whitney U statistical test (**** $p \leq 0.0001$; *** $p \leq 0.001$). All associated *p*-values are presented in the S1 Table.

Deletion of *RAD52* reduces the instability of $(GT)_{49}$ in the wild-type background by 70% (Fig 5B). The absence of *RAD52* also strongly reduces the increased instability of the $(GT)_{49}$ tract by 75%, in the *psf1-1* mutant (Fig 5B). For the $(AACGCAATGCG)_4$ and $(CAACGCAATGCGTTG GATCT)_3$ tracts, deletion of *RAD52* in the wild-type background reduces the instability by 60% and 35%, respectively (Fig 5C and 5D). In the double *psf1-1 rad52Δ* mutant strain, the instability of these tracts is reduced by 82% and 66%, respectively, compared to that in the *psf1-1* strain (Fig 5C and 5D). These data indicate that Rad52-dependent recombination events are closely associated with the instability of the tested repeat tracts in the *psf1-1* background.

The central and highly conserved step in HR is the formation of Rad51 nucleoprotein filaments on the single-stranded DNA ends. Among its many functions in DNA replication and repair, Rad51 is crucial for the homology search and strand invasion steps [96–99]. Therefore, loss of this important player in most of the pathways of HR could be expected to affect DNA repeat tract instability in the *psf1-1* strain. In the wild-type strain with the random control sequence, deletion of *RAD51* results in an almost 8-fold increase in *URA3* mutagenesis (Fig 5E). The *psf1-1* and *rad51Δ* mutations together lead to a further additive increase in forward mutagenesis compared to the single mutants (Fig 5E). Based on the observed additive effect of *psf1-1* mutation with *RAD52* or *RAD51* deletion we conclude that recombination has no significant contribution to forward mutagenesis observed in *psf1-1* cells. In the wild-type strain, deletion of *RAD51* reduces the instability of the repeat tracts $(GT)_{49}$, $(AACGCAATGCG)_4$ and $(CAACGCAATGCGTTGGATCT)_3$ by 79%, 47% and 33%, respectively (Fig 5F–5H). Inactivation of *RAD51* in the *psf1-1* background reduces the instability of these tracts by 77%, 67%, and 55%, respectively, compared to that in the *psf1-1* strain (Fig 5F–5H). These results confirm that the instability of repeated sequences in *psf1-1* cells is highly dependent on recombination.

When HR is activated, the requirement for Rad51 nucleoprotein filaments increases, so the Rad51 level is elevated. Since in *psf1-1* cells, the stability of the repetitive sequences depends on HR, we asked whether the Rad51 level is elevated in this mutant. Indeed, as shown in Fig 6A and 6B, the Rad51 protein level is 1.5-fold higher in the *psf1-1* mutant than in the wild-type strain.

## Accumulation of ssDNA in *psf1-1* cells

HR substrates are DSBs or ssDNA stretches that can be formed during impaired DNA replication, especially in cells with defective replisomes. The observed recombination-dependent increased instability of DNA repeat tracts in *psf1-1* cells suggested that ssDNA regions accumulate in this mutant. To verify this hypothesis, we embedded yeast chromosomes in agarose plugs and treated the plugs with S1 nuclease, which cleaves dsDNA at single-stranded regions such as nicks, gaps, or loops. Next, we visualized the integrity of the chromosomes by separating them using pulsed-field gel electrophoresis (PFGE). In addition to the wild-type and *psf1-1* strains, we also used the *pol32Δ* strain, which accumulates ssDNA gaps [100], as a control. In the *psf1-1* and *pol32Δ* strains, we observed strong degradation of chromosomes after S1 nuclease treatment compared to that in the wild-type strain (Fig 6C). This result is consistent with the idea that ssDNA stretches accumulate in *psf1-1* cells. Single-stranded DNA is coated by Replication Protein A (RPA). Therefore, to estimate the degree of ssDNA formation in *psf1-1* cells, we analyzed the formation of RPA-bound ssDNA through visualization of its subunit Rfa1 fused with YFP. We observed an increased number and higher intensity of Rfa1-YFP foci in *psf1-1* cells compared to that in wild-type cells (Fig 6D and 6E).

## Mms2 influences the stability of repeat tracts in *psf1-1* cells

The recombination proteins Rad52 and Rad51 are also involved in template switching (TS), a recombination-related pathway that enables survival of fork stalling, DNA lesions, or

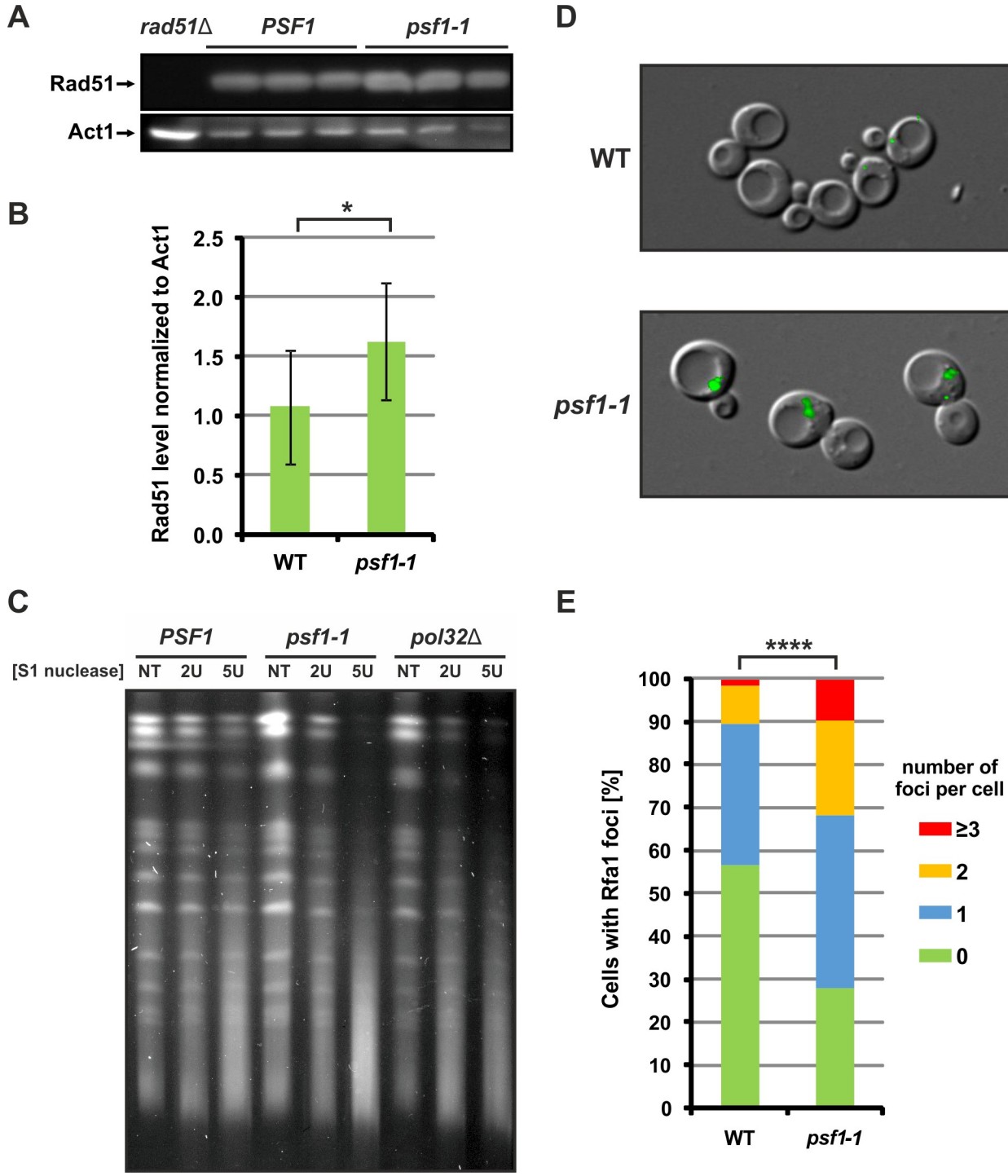

**Fig 6. Accumulation of single-stranded DNA and activation of the HR pathway in *psf1-1* cells. (A)** Rad51 protein levels in the wild-type and *psf1-1* strains were analyzed by western blotting with an anti-Rad51 antibody. **(B)** Quantification of the western blotting results showing the mean ±s.d. of 8 to 12 repetitions of the assay. Strain BY4741 *rad51Δ* served as a negative control. Statistical significance was determined with Student's t-test (*p*-value * ≤0.05) **(C)** Pulsed-field gel electrophoresis (PFGE) analysis of chromosome integrity after S1 nuclease treatment of DNA from yeast cells cultured at the permissive temperature 23˚C. Agarose plugs with DNA were treated with two units (2 U) or five units (5 U) of S1 nuclease for 30 min. Untreated plugs are shown as a control (NT). Strain *pol32Δ* served as a positive control. **(D)** Rfa1–YFP foci detected in wild-type and *psf1-1* strains. **(E)** Rfa1 foci frequency quantification. Three biological replicates were performed, each with at least 200 cells counted. The results represent the number of cells with indicated number of foci with SD. For statistical analysis contingency table and the chi-square test were used (S2 Table). The chi-square statistic is 223.4038, which corresponds to the *p*-value ****<0.0001.

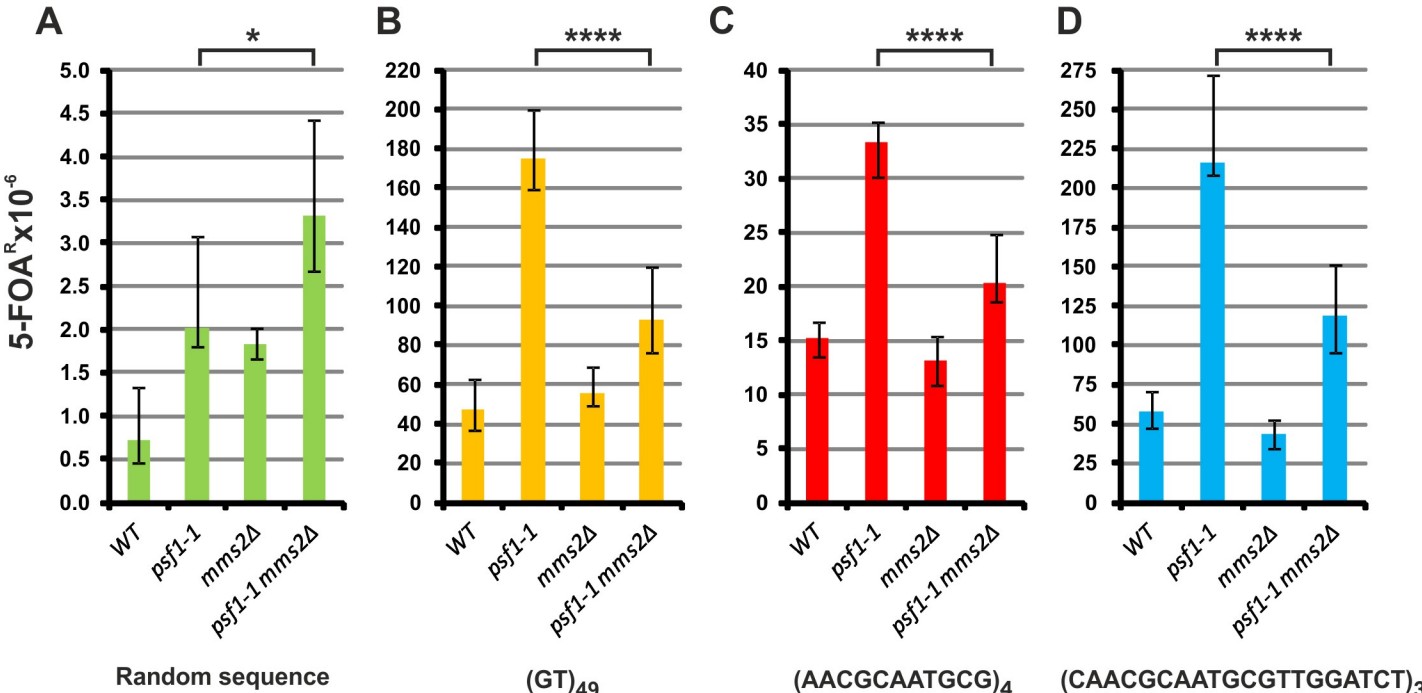

**Fig 7. Template switch mechanism contributes to the instability of repeat tracts in *psf1-1* cells (A-D).** The 5-FOA$^R$ mutation rates were determined at 23˚C for yeast cells with the indicated genotypes carrying plasmids with the analyzed sequences; the values are medians with 95% confidence intervals calculated from data for at least ten cultures of each strain; the *p* values for *psf1-1 mms2Δ* versus *psf1-1* strains were calculated using the nonparametric Mann-Whitney U statistical test (* *p*≤0.05; **** *p*≤0.0001). All associated *p*-values are presented in the S1 Table.

impediments [93,101,102]. This process uses the newly synthesized homologous daughter strand as a template to enable the continuation of DNA replication. TS requires further ubiquitylation of the monoubiquitinated proliferating cell nuclear antigen (PCNA) in a process driven by the Ubc13–Mms2–Rad5 E2–E3 polyubiquitylating complex [103,104]. Inactivation of Mms2 has an effect on forward *URA3* mutagenesis in both wild type and *psf1-1* cells (2.5-fold and 1.6-fold increase, respectively) (Fig 7A). The additive effect of *psf1-1* mutation and *MMS2* deletion demonstrate that TS does not contribute to the enhanced mutagenesis in *psf1-1* cells. *MMS2* deletion has no effect on the instability of repeat tracts in the wild-type strain (Fig 7B–7D). However, in *psf1-1* cells, inactivation of *MMS2* significantly reduces the instability of (GT)$_{49}$, (AACGCAATGCG)$_4$ and (CAACGCAATGCGTTGGATCT)$_3$ tracts by 47%, 39%, and 45%, respectively. This finding indicates that TS may be involved in the instability of repetitive sequences in *psf1-1* cells.

## Break-induced replication is involved in the instability of repeated sequences in *psf1-1* cells

Another subpathway of HR is break-induced replication (BIR) [105]. This process, when involved in the synthesis of repeated DNA sequences leads to an elevated risk of genetic instability [106]. Therefore, we decided to test whether BIR may be involved in the instability of repeated sequences in the *psf1-1* mutant by deleting genes encoding proteins involved in this process. Pol32, is involved in BIR not only as a nonessential subunit of Pol δ, but also contributes to the establishment of a repair replication fork, i.e., strand displacement during bubble migration [107–110]. Therefore it is specifically involved in BIR but not in other HR-related mechanisms. The Pol32 subunit is not essential for normal DNA replication; however, deletion

of *POL32* causes cold sensitivity at 13˚C [111]. The *psf1-1* strains also exhibits temperature sensitivity and do not grow at 37˚C. We attempted to construct *psf1-1 pol32Δ* strains and obtained double mutants viable at 28˚C that exhibit growth impairment at 15˚C and 37˚C. Deletion of *POL32* has no significant effect on forward mutagenesis in *URA3* in the wild-type strain (Fig 8A). Pol32 is one of two subunits shared by two DNA polymerases: Pol ζ and Pol δ. Because Pol ζ participates in mutagenesis in both wild-type and *psf1-1* strains, one can expect that deletion of *POL32* in these strains will reduce forward mutagenesis. However, similar to previously published data, our results show that this is not the case [112,113]. The lack of expected mutagenesis decrease is probably associated with the fact that Pol32 not only plays a role as a Pol ζ subunit but also participates in Pol δ activities. Therefore, in the *pol32Δ* strain, the antimutator effect on Pol ζ-dependent mutagenesis is compensated by a moderate mutator effect associated with the involvement of Pol32 in the reactions of Pol δ. For this reason, the effect of *pol32Δ* is moderate in the *psf1-1* background (Fig 8A).

Deletion of *POL32* reduces the instability of $(GT)_{49}$ tracts both in wild-type and *psf1-1* cells by 84% and 92%, respectively (Fig 8B). For the $(AACGCAATGCG)_4$ and $(CAACGCAATGC GTTGGATCT)_3$ tracts, inactivation of *POL32* had no effect on tract instability in wild-type cells (Fig 8C and 8D) but reduces the instability in the *psf1-1* background by 29% and 62%, respectively (Fig 8C and 8D). These results suggest that the mechanisms responsible for the instability are not the same for the microsatellite $(GT)_{49}$ tracts and minisatellite $(AACGCAA TGCG)_4$ and $(CAACGCAATGCGTTGGATCT)_3$ tracts.

To extend our investigation of the involvement of BIR in repeated sequence instability, we constructed *pif1Δ* strain derivatives. *PIF1* encodes a DNA helicase involved in various DNA transactions, including BIR, or maturation of Okazaki fragments [108,114–116]. Deletion of *PIF1* has a significant effect on forward mutagenesis in wild-type cells (6.4-fold increase) probably due to inactivation of mitochondrial activity (loss of mtDNA, $rho^0/rho^-$) [117–119] leading to increased Pol ζ-dependent nuclear mutagenesis [120]. The observed similar mutagenesis levels in *pif1Δ* and *pif1Δ psf1-1* cells (Fig 8E) can be explained by the observation that participation of Pol ζ in DNA replication was already increased in the GINS mutant (Fig 4A). We also observed that inactivation of *PIF1* has no effect on the stability of the tested repeat tracts compared to that in the wild-type strain (Fig 8F–8H). However, analysis of the genetic interactions between *psf1-1* and *pif1Δ* shows that a lack of functional Pif1 helicase decreases the instability of the $(GT)_{49}$, $(AACGCAATGCG)_4$ and $(CAACGCAATGCGTTGGATCT)_3$ repeated sequences by 58%, 29%, and 53%, respectively. Together, the results obtained for the *pol32* or *pif1* deletion mutants indicate that BIR contributes to DNA repeat instability in the *psf1-1* strain.

## Discussion

In this work, we investigated the impact of the proper functioning of GINS on the stability of repeated DNA sequences. GINS is a component of one of the crucial leading-strand replication elements of the replisome, i.e., the CMG-E complex of DNA helicase (Cdc45, Mcm2-7, GINS) with Pol ε [57]. Although the structural dynamics of this large complex have been extensively analyzed [121,122], the physiological role of the individual subunits is poorly understood, especially their role in the maintenance of genomic stability. In addition to its essential role in DNA unwinding, CMG is a platform that functionally and structurally coordinates the participation of various replisome elements in DNA replication. Previously, we showed that the Psf1-1 mutant form of the GINS subunit increases the levels of both base substitutions and indels [62]. Here, we show that the *psf1-1* allele increases the instability of repeat tracts located on plasmids and various chromosomally encoded trinucleotide repeats. Of particular interest is

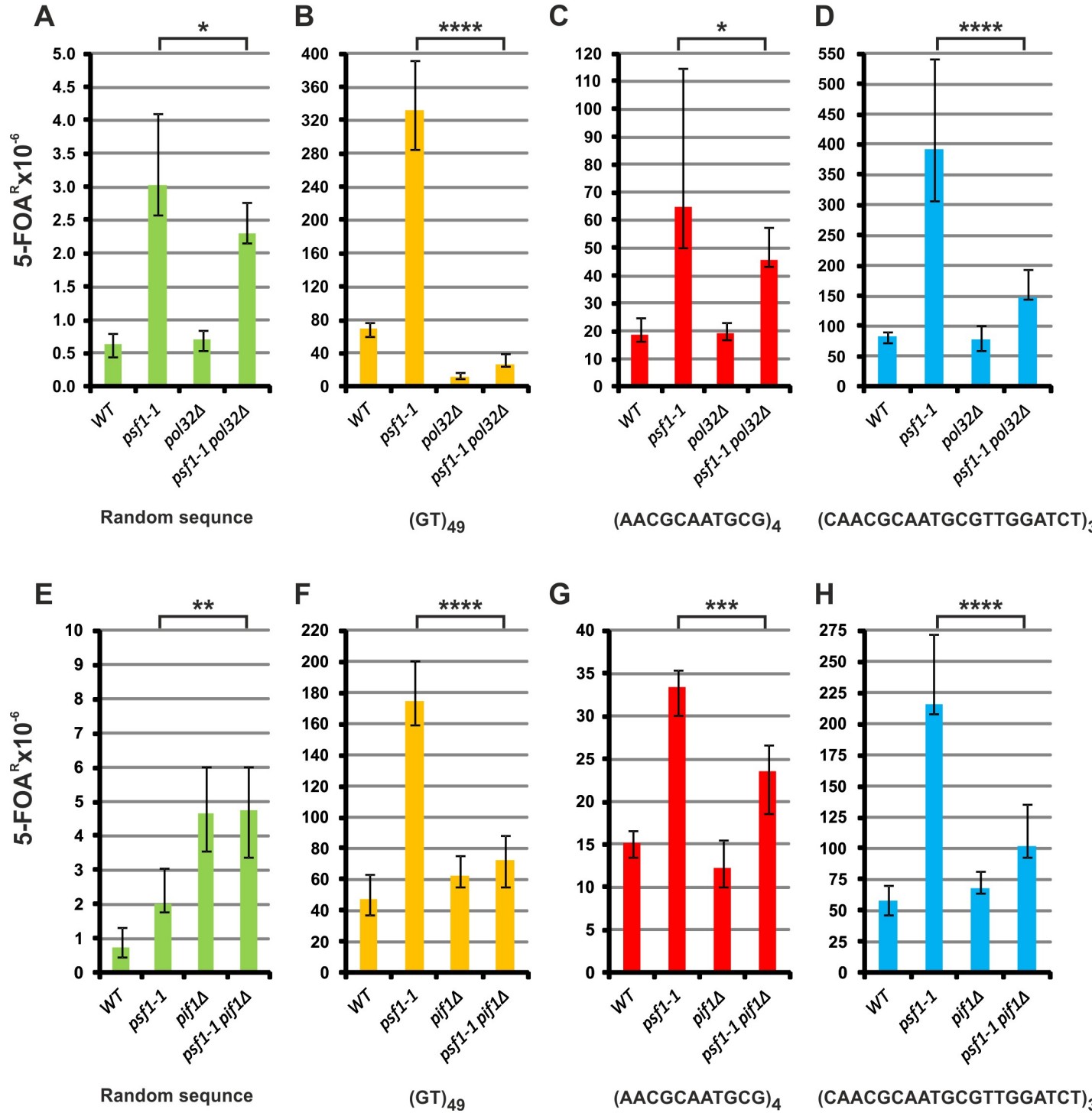

**Fig 8. Break-induced replication contributes to the instability of repeat tracts in *psf1-1* cells. (A-D) Impact of *POL32* disruption. (E-H) Impact of *PIF1* disruption.** The 5-FOA$^R$ mutation rates were determined at 28°C for yeast cells with the indicated genotypes carrying plasmids with the analyzed sequences; the values are medians with 95% confidence intervals calculated from data for at least ten cultures of each strain; the *p* values for *psf1-1 pol32Δ* or *psf1-1 pif1Δ* versus *psf1-1* mutants versus wild-type strains were calculated using the nonparametric Mann-Whitney U statistical test (**** $p \leq 0.0001$; *** $p \leq 0.001$; ** $p \leq 0.01$; * $p \leq 0.05$). All associated *p*-values are presented in the S1 Table.

the observed Psf1-1-dependent instability of GAA and TTC repeats, expansions of which are involved in Friedreich ataxia in humans [123]. In the wild-type background, expansions of GAA and TTC are not frequent unless the number of repeats exceeds a length threshold (78 triplets) [124]. Additionally, the repeated CTG sequences, the instability of which is the cause of human disorders such as Huntington disease, myotonic dystrophy and spinocerebellar ataxia, also exhibit significantly increased instability in *psf1-1* cells (Fig 1). It was demonstrated that trinucleotide repeats can form structures that result in stalling of DNA synthesis *in vitro* [125–127] and/or impede replication fork progression in vivo [128–131]. Based on the results for *psf1-1*, we conclude that functional impairment of CMG-E can enhance these destabilizing effects.

Why is proper functioning of a noncatalytic subunit of the GINS complex so significant for the stability of repetitive sequences? Answering this important basic research question will provide a better understanding of the molecular processes that lead to the DNA repeat instability that has been observed in many human diseases. As the Psf1-1 protein exhibits a greatly impaired interaction with the Psf3 subunit of the GINS complex [54,62], this mutant may influence the integrity of the GINS complex, affecting communication with other components of CMG, Pol ε and/or DNA.

What might be the mechanisms involved in DNA repeat instability in *psf1-1* cells? Although deletion of the *REV3* gene, which inactivates Pol ζ, reduces by 40–50% the rate of mutagenesis in the *CAN1* [62] and *URA3* reporter gene (Fig 4A), inactivation of Pol ζ does not affect the stability of repeat tract sequences (Fig 4B–4D). This result is consistent with previous studies showing that defects in trans-lesion synthesis polymerases do not influence repeat instability in wild-type budding yeast [44,132,133]. However, in cells with impeded replicative polymerases, some repeat expansions do occur *via* a Polζ-dependent mechanism [44], as do short duplications initiated by small hairpins [134]. Nevertheless, our results indicate that mutation in a subunit of GINS, a replisome component, results in Pol ζ-independent repeat instability.

Previous studies on mechanisms of microsatellite instability as well as analyses of mutations generated *via in vitro* replication assays demonstrated that changes in the number of repeat tracts often result from several mechanisms, e.g., formation of non-B DNA structures, strand slippage, reduced processivity or dissociation of DNA polymerase [67,74,135–138]. Some errors, such as base substitutions or small DNA loops, that arise during replication may be the target for the MMR system [127,139]. We showed that MMR corrects a majority of the spontaneous errors produced in *psf1-1* strains (Fig 3A), [62]. The statistically significant increase in polyGT instability in the *psf1-1 msh2Δ* mutant compared to that in the single mutants indicates that the *psf1-1* allele enhances the frequency of MMR-corrected microsatellite instability. However, inactivation of *MSH2* has no effect on the instability of long minisatellite sequences in *psf1-1* cells, which is consistent with the specificity of loop-size recognition by the MMR system.

Another mechanism that is frequently involved in the instability of repeated DNA sequences is DNA recombination—for a review, see [27]. This process is involved in the reactivation of stalled or collapsed replication forks and in the repair of DNA gaps left behind the replication fork which might arise in the *psf1-1* cells as suggested by accumulation of increased number of single-stranded DNA regions and Rfa1 foci (Fig 6C–6E) shown in this work. HR, which is generally considered to be an error-free process, may be an important source of genomic instability due to rearrangements between the invading and homologous strands with repetitive regions, formation of hairpins during D-loop extension, and difficulties associated with synthesis across repetitive regions, especially those that form DNA secondary structures [27,140–142]. The most important recombination proteins are Rad52 and Rad51 [96,98,143]. Deletion of *RAD52* or *RAD51* reduced the instability of repeat tracts compared to the *psf1-1*

mutant (Fig 5B–5D) what supports the conclusion that HR mechanisms are involved in this process (Fig 5F–5H). An additional fact reinforcing this conclusion is the elevated level of Rad51 in *psf1-1* cells (Fig 6A and 6B). It seems that an increased level of Rad51 is the supportive mechanism for cells during replication slow-down or blockage. A similar effect was documented for cells lacking the transcription factor Swi6, which controls the expression of G1/S transition genes, resulting in limitation of proteins involved in DNA replication and repair and a prolonged cell cycle. Their survival is enhanced by Rad51-dependent illegitimate recombination [93]. Similar phenotypes were also described for human cells, in which mutation in the KRAS proto-oncogene caused replication fork stalling, DNA lesion accumulation and increased abundance of various proteins, including Rad51, which is obligatory for the survival of these mutant cells [144].

To further analyze the recombination-dependent mechanisms involved in *psf1-1*-mediated instability of repeat tracts, we inactivated the template switch mechanism, which depends on the activity of the Ubc13-Mms2-Rad5 complex [102–104,145,146]. The observed decrease in instability for all the tested repeated sequences in the *psf1-1 mms2Δ* strains (Fig 7B–7D) indicates that the Mms2-dependent template switch mechanism may be involved in the instability of repeat tracts in the *psf1-1* mutant.

Another possible mechanism that enables the continuation of DNA synthesis after replication fork stalling or collapse is BIR, in which the main proteins involved are Pol δ with its Pol32 subunit as well as the Pif1 helicase [108,114,116]. The involvement of BIR in the generation of large-scale expansions was shown previously [106,147]. Here, we show that deletion of *POL32* or *PIF1* significantly decreases the high instability of the tested tracts in the *psf1-1* background (Fig 8B–8D and 8F–8H) suggesting that BIR may be one of the mechanisms underlying the formation of repeated sequence instability in the *psf1-1* strains.

van Pel and colleagues [148] showed that the *psf1-1* mutant exhibits a dramatic increase in Rad52 foci frequency and a two-fold increase in the number of G2/M arrest large-budded cells. These data together with previously showed retarded S-phase progression [54], dumbbell-cell formation and impaired interaction within the GINS complex and genomic instability [62], allow us to conclude that Psf1-1 significantly impairs the replication process. It should be stressed here that the observed *psf1-1* phenotypes may result from a combination of processes activated in response to defective DNA replication and these mechanisms may be envisioned as sources of increased instability of repetitive DNA tracts in the *psf1-1* strains.

First, impaired replication and accumulation of ssDNA gaps may promote the formation of non-B DNA secondary structures by repeated sequences. Such structures formed on the nascent strand will promote expansions, whereas contractions may result from polymerase stalling before a hard-to-replicate hairpin structure followed by 3′ end displacement to a new position within the repeated sequence template–for a review, see [27].

Second, impaired replication due to the low-stability CMG-E complex may promote a wide range of events, such as enhanced DNA / polymerase slippage, disruption of the helicase-polymerase interaction, uncoupling of leading and lagging DNA strand syntheses, fork restart and/or enhanced repriming. The mechanism allowing repriming on the leading DNA strand and recycling of stalled leading strand polymerase for downstream synthesis remains under extensive investigation. This mechanism was shown to be primase-coupled in bacteria [149,150] and helicase-coupled in yeast [151]. Interestingly, Hashimoto and colleagues showed that upon fork collapse, the active Cdc45–Mcm2-7–GINS (CMG) helicase complex loses its GINS subunit and interaction with Pol ε [152]. Then, Rad51 and Mre11 are required for functional replisome reassembly and reloading *via* a recombination-mediated process. In *psf1-1*, the need for recombination-dependent re-establishment of the replisome can be enhanced if the defects in GINS increase the uncoupling of GINS from the replisome.

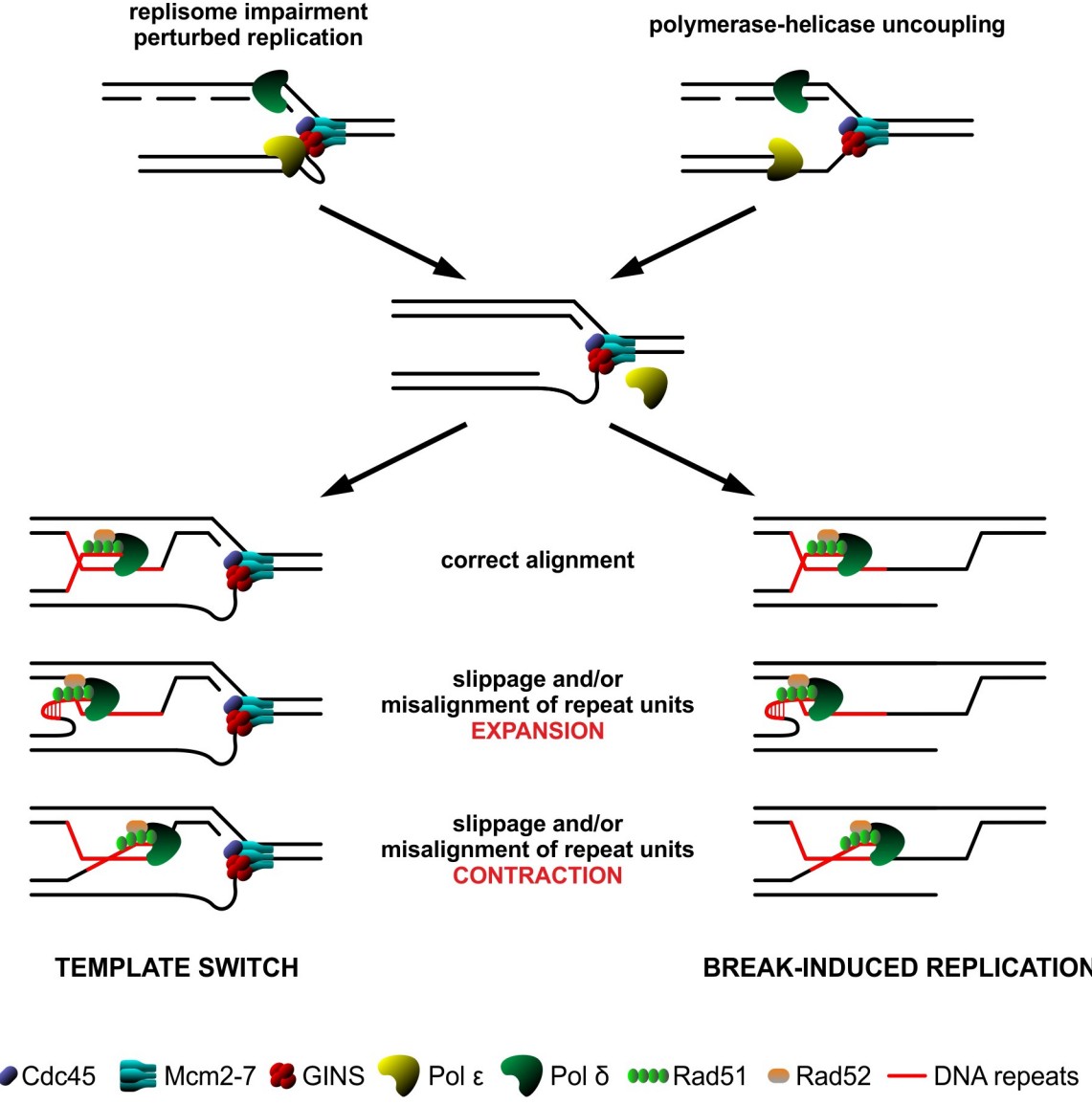

**Fig 9. Schematic model for the DNA repeats instability mechanisms in the *psf1-1* mutant.** Defective interactions within the GINS complex of CMG-E may lead to replication perturbations and/or polymerase-helicase uncoupling and, as a consequence, formation of ssDNA gaps. To ensure the continuity of DNA duplication cells can employ various mechanisms including template switch (TS) and break-induced replication (BIR). These mechanisms require homology search which in case of a repeated DNA tract may result in misalignment leading to its expansion or contraction. Moreover, such DNA sequences may be prone to replication slippage enhancing their instability.

Finally, the increased amounts of ssDNA and Rfa1 foci (Fig 6C–6E) suggest that in the *psf1-1* cells, Pol ε and the CMG helicase may be uncoupled. Nicks or small gaps that arise during impaired DNA replication can be enlarged. Misalignments occurring during homology search between the invading and the template DNA strand may promote expansions and/or contractions. Nicks and gaps can also become DSBs if replicated or if they occur as a result of breakage due to the intrinsic fragility of ssDNA [153–155] serving as substrates for Rad51 and Rad52 to initiate recombination and enhance DNA repeat instability during D-loop extension (Fig 9). Importantly, as mentioned previously, the *psf1-1* mutant exhibits a dramatic increase in Rad52 foci [148]. These observations are consistent with the reported strong association between faulty replication-induced tract instability (i.e., slippage during synthesis, replication fork

pausing) and recombination [27,68,156]. The question of how different recombination pathways are preferentially activated over others and how the transition between different modes of action occurs requires further investigation. Nonetheless, our present results provide new insights into the source of repetitive sequence instability.

These findings shed new light on the impact of the GINS complex on genome stability in the context of recently described disorders in which mutations or deregulation of the human homolog of the *PSF1* gene was observed [60,157]. Importantly, our results expand the list of genes that, when mutated, affect the structures and functions of other genes and structural elements of the chromosome that contain repetitive DNA sequences.

## Materials and methods

### Strains, media, and growth conditions

*Escherichia coli* DH5α (F⁻, *gyrA96*, *recA1*, *relA1*, *endA1*, *thi1*, *hsdR17*, *supE44*, *deoR*, Δ(lacZYA-argF)U169, [φ80Δ (*lacZ*)M15]) was used for cloning and plasmid amplification. Bacterial strains were grown at 37°C in LB medium (liquid or solidified with 1.5% agar), supplemented when needed with ampicillin 100 μg/ml (Polfa Tarchomin S.A., Warsaw, Poland). The *Saccharomyces cerevisiae* strains were grown at 23 or 28°C in standard media [158,159]. YPD complete medium (1% Bacto-yeast extract, 2% Bacto-peptone, 2% glucose) was used when nutrient selection was not required. Transformants were selected on YPD with appropriate antibiotics: hygromycin B 300 μg/ml (Bioshop, Burlington, Canada) or nourseothricin 100 μg/ml (Werner BioAgents, Jena, Germany). Yeast strains were selected for prototrophy on synthetic SD minimal medium (0.67% yeast nitrogen base without amino acids, 2% glucose) supplemented with appropriate amino acids and nitrogenous bases. Solid media were obtained by the addition of 2% agar. SD medium with 5-FOA 1 mg/ml (USBiologicals, Salem, MA, USA) was used for the selection of *URA3* mutant cells [72].

### General methods

Yeast strains were transformed using the LiAc/ssDNA/PEG method [160]. Total DNA was isolated from yeast cultures using the Genomic Mini AX Yeast Spin Kit (A&A Biotechnology, Gdansk, Poland). *E. coli* cells were transformed as described by Sambrook and Russell [161]. Plasmids were isolated from bacteria using the Plasmid Mini Kit (A&A Biotechnology). DNA was extracted from the agarose gel using the Gel-Out Kit (A&A Biotechnology) and purified after enzymatic reactions using the Clean-Up Kit (A&A Biotechnology). Restriction enzymes and DNA ligase (Thermo Scientific, Waltham, MA, USA) were used as recommended by the supplier.

### Construction of yeast strains

Yeast strains used in this work are the derivatives of ΔI(-2)I-7B-YUNI300 [162], detailed in S3 Table. Strains carrying deletion of the *REV3*, *RAD51*, *RAD52*, *MSH2*, *MMS2*, *PIF1* or *POL32* gene were constructed based on the SC801 strain [62] by replacing the coding region of the appropriate gene with a DNA cassette containing the *HPH* or *NAT1* gene, which was PCR-amplified with the primers listed in S4 Table using pAG32 or pAG25 [163] as a template. Strains carrying the same deletions and the *psf1-1* allele were constructed by tetrad dissection from diploid strains constructed by crossing strain SC802 with appropriate single gene deletion MATα strains listed in the S3 Table. Gene replacement was confirmed by PCR using primers listed in S4 Table. The presence of *PSF1* or the *psf1-1* allele was verified by a temperature sensitivity test (*psf1-1* strain does not grow at 37°C) and PCR (primers: InProm, dwPSF1, listed in S4 Table) followed by DNA sequencing.

Strains carrying trinucleotide repeat tracts or the control random sequence (S5 Table) were prepared by integration of the plasmids pMA1 (25xCTG), pMA5 (25xGAA), pMA6 (25xTTC), and pMA7 (control sequence), (see S5 Table) into the SC801 (*PSF1*) or SC802 (*psf1-1*) strain. The integrative plasmid pMA1 was constructed as follows: to replace the *HIS3* selectable marker with *TRP1*, a 5735-bp PfoI-PsiI fragment of the pBL69 plasmid (kindly provided by R. Lahue [65]) was ligated with a 1287-bp PfoI-PsiI fragment of the pRS314 plasmid (ATTC, Manassas, Virginia, US). Other integrative plasmids were constructed by replacing the repeat tract from pMA1 with the appropriate insert obtained by SphI digestion of oligonucleotide duplexes: pMA5: 5'GCGCGCGCATGCGAAGAAGAAGAAGAAGAAGAAGAAGA AGAAGAAGAAGAAGAAGAAGAAGAAGAAGAAGAAGAAGAAGAAGAAGAAGAAGCATG CCGCGCG3', pMA6: 5'CGCGCGGCATGCTTCTTCTTCTTCTTCTTCTTCTTCTTCT TCTTCTTCTTCTTCTTCTTCTTCTTCTTCTTCTTCTTCTTCGCATGCGCGCGC3', and pMA7: 5'CGCGCGGCATGCCCAGGTCGCCGTCGTCCCCGTACGCGACGAACG TCCGGGAGTCCGGGTCGCCGTCCTCCCCGTCGTCCGATTCGCATGCGCGCGC3'. The obtained plasmids were linearized with BlpI and integrated into the *LYS2* locus of strains SC801 and SC802, which was confirmed by PCR using primers Lys2A and Lys2D, listed in S4 Table. To exclude possible integration into the *TRP1* or *URA3* locus, PCR was performed with the primers TRP1A, TRP1D, URA3UP, and URA3LW, listed in S4 Table. The appropriate length of each tract or control random sequence was verified by PCR (primers: OBL157 and Tri1S, listed in S4 Table) followed by DNA sequencing.

The *RFA1-YFP* fusion was introduced by yeast transformation with the *RFA1-YFP LEU2* cassette amplified using primers RFA7317F and RFA6231R from the plasmid pRYL24. To construct this plasmid, primers RFA7317F and RFA6231R were used to amplify the *RFA1-YFP* cassette from the W3775-12C chromosome and cloned into the SmaI-digested vector pMT5 (*ori*$_{pMB1}$, Tc$^R$) [164] propagated in *E. coli*. Then, *LEU2* was amplified with primers LEU2_F_H and LEU2_R_H from the plasmid pRS315 and cloned into a ScaI site downstream of *RFA1-YFP* to obtain the plasmid pRYL24.

## Stability of trinucleotide repeat tracts integrated into the chromosome

The assay used in this study is a modification of the system developed by R. Lahue's group [165]. Single colonies of yeast strains were used to inoculate 2 or 20 ml (depending on the expected mutagenesis rates) of SD medium supplemented with the required amino acids and nitrogenous bases, lacking tryptophan and leucine (10–30 cultures for each strain) and in parallel were tested for repeat tract or control sequence length by colony PCR (with the primers OBL157 and Tri1S, listed in S4 Table) and 2% agarose gel electrophoresis. Then, after 72 h of incubation at 23˚C, 10–30 cultures of each strain were appropriately diluted and plated on selective (containing 5-FOA for selection of *URA3* mutants) and nonselective (without 5-FOA) media. Colonies were counted after 4–7 days of incubation at 23˚C. The spontaneous mutation rates were determined as described below.

## Stability of repetitive sequences located on plasmids

To evaluate the level of repetitive sequence instability, the following set of plasmids containing the origin of replication from ARS1 was used: pMD28 (18x1 nt), p99GT (49x2 nt), pMD41 (4x11 nt), and pEAS20 (3x20 nt) (kindly provided by T. Petes). In each plasmid, the repetitive sequence is inserted upstream of *URA3* gene (fused with a small region of the *HIS4* gene), in-frame with its coding sequence, as described previously [66,166]. The control plasmid pKK2 was created by inserting the control sequence (S5 Table) obtained by SalI-XhoI digestion of the oligonucleotide duplex, 5'GTCGACATGCGCTGGCCGCTTGCGTTGCGTCGTTGCT

CTTTCTCGAG3', into the SalI-XhoI-digested plasmid pSH44 [66]. The presence of the control sequence in the constructed plasmid was verified by PCR (with the primers GT_FOR and GT_REV, listed in S4 Table) followed by DNA sequencing. These plasmids were used for the transformation of yeast strains SC801, SC802 and their derivatives carrying deletions of the appropriate genes (S3 Table).

To determine mutation rates, 8–20 cultures of 2 or 3 independent isolates of each strain were inoculated in 2 ml of liquid SD medium supplemented with the required amino acids and nitrogenous bases, lacking tryptophan and leucine, and grown at 23°C. In the experiment with the *psf1-1 pol32Δ* mutant, all strains were grown at 28°C. When cultures reached the stationary phase, appropriate dilutions were plated on selective (containing 5-FOA for selection of *URA3* mutants) and nonselective media. Colonies were counted after 4–7 days of incubation at 23 or 28°C. The spontaneous mutation rates were determined as described below.

To define the spectrum of changes within the repeat tracts or the control sequences, total DNA from 47–85 5-FOA-resistant colonies from independent cultures of each strain was isolated and used for PCR amplification of the analyzed region with fluorescently labeled primers: REP1_FAM/ REP2_ROX/ REP3_HEX/ REP4_TAMRA and GT_FOR (listed in S4 Table). Changes in sequence length were identified by capillary electrophoresis and analyzed using Peak Scanner software v1.0.

## Determination of spontaneous mutation rates

The mutation rates were calculated using the equation $\mu = f/ln(N\mu)$, where $\mu$ is the mutation rate per round of DNA replication; *f* is the mutant frequency (cell count from selective media divided by the cell count from nonselective media), and *N* is the total population size [167]. The median values of the mutation rates and 95% confidence intervals were evaluated with STATISTICA 6.0 software. Statistical significance of differences in the mutation rates between the respective strains (*p*-values) was measured using the nonparametric Mann-Whitney *U* test while Kruskal-Wallis test with the posthoc Mann-Whitney U-test with Bonferroni correction for multiple comparisons was applied to analyze rates in three or more individual cultures.

## Visualization of ssDNA in yeast chromosomes by S1 nuclease treatment and separation by pulsed-field gel electrophoresis

Yeast chromosome integrity was analyzed as described previously [168] with certain modifications. Yeast cells grown at 23°C were embedded in 20-μl plugs of low-melting-point SeaKem Gold agarose (Lonza, Basel, Switzerland). The plugs were digested with Zymolyase 100T (Bio-Shop) overnight at 37°C with gentle rotation and then with proteinase K (A&A Biotechnology) and RNase A (Sigma-Aldrich, St. Louis, MO, USA) overnight at 37°C with gentle rotation. Then, the plugs were treated with 2 U or 5 U of S1 nuclease (Thermo Scientific) for 30 min in S1 buffer provided by the manufacturer. Next, plugs were placed in the wells of a 1% D5 agarose gel (BioMaxima, Lublin, Poland) in 1x TAE and sealed with the same agarose. Electrophoresis was performed on a CHEF Mapper XA pulsed-field electrophoresis system (Bio-Rad, Hercules, CA, USA) for 18 h in 1x TAE buffer at 6 V/cm and 12°C, angle of 120°, and switch time of 60–85 s, with a ramp-up of 0.8. After electrophoresis, the DNA was stained with 0.5 μg/ml ethidium bromide (Sigma-Aldrich) and visualized using UV light.

## Determination of Rad51 protein levels by western blot

For western blotting, $1\times10^8$ cells from exponentially growing liquid culture (density $5\times10^6$ ml$^{-1}$) were collected by centrifugation. Protein extracts prepared using the TCA method were suspended in Laemmli sample buffer supplemented with 1 mM PMSF and cOmplete Protease

Inhibitor Cocktail (Roche, Base, Switzerland), and boiled for 5 min. After centrifugation (19,300 g for 2 min), equal volumes of the protein extracts were separated by SDS-PAGE (8% polyacrylamide gel), and the proteins were transferred onto PVDF membrane (GE Healthcare, Pittsburgh, PA, USA). Blots were blocked for 2 h in 5% (w/v) nonfat dried milk before anti-Rad51 detection or in 3% (w/v) BSA before anti-Act1 detection; both were dissolved in TBST [25 mM Tris-HCl pH 7.5, 137 mM NaCl, 27 mM KCl, 0.1% (v/v) Tween-20]. Rad51 protein was detected with the rabbit polyclonal antibody anti-Rad51 (1:2000, Thermo Fisher, PA5-34905) and goat anti-rabbit IgG conjugated to horseradish peroxidase (HRP) (1:2000, DAKO, P0448). Actin was detected by using a mouse anti-actin monoclonal antibody (1:3000, C4, CHEMICON, MAB-1501) and goat anti-mouse IgG (H+L) alkaline phosphatase (AP)-conjugated (1:3000, Bio-Rad, 1706520) antibody. Immunoreactive proteins on the blots were visualized using chemiluminescent substrates: SuperSignal WestPico, PIERCE for HRP and CDP-Star, ready-to-use, Roche for AP and documented with a charge-coupled device camera (FluorChem Q Multi Image III, Alpha Innotech, San Leandro, CA). The resulting bands were quantified by using Image Quant 5.2 (Molecular Dynamics, Inc., Sunnyvale, CA). The protein levels were normalized to those of Act1. The results of at least eight biological repetitions were averaged to determine the relative protein levels. Statistical significance was determined with Student's t-test.

### Fluorescence microscopy

The yeast strains containing the chromosomal fusion of *RFA1-YFP* were grown at 23˚C to exponential phase in SD medium supplemented with the required amino acids and nitrogenous bases. The Rfa1-YFP foci were examined using a Nikon E800 fluorescence microscope with a FITC filter (EX465-495, BA 515–555). Images were processed with ImageJ software [169]. At least 200 cells were screened for each of three biological repeats. To analyze the possible differences in Rfa1-Yfp foci in wild-type and *psf1-1* strains we used the contingency tables and further applied the chi-square test (S2 Table).

## Supporting information

**S1 Fig. Predicted structures formed by repeat tracts. (A)** (AACGCAATGCG)$_4$ and **(B)** (CAACGCAATGCGTTGGATCT)$_3$. Predictions were made using the RNAstructure web server for nucleic acid secondary structure prediction (https://rna.urmc.rochester.edu/RNAstructureWeb/Servers/Predict1/Predict1.html).
(TIF)

**S1 Table. *p*-values associated with data presented in Figs 1, 2, 3, 4, 5, 7 and 8.**
(PDF)

**S2 Table. Statistical analysis of Rfa1 foci in Wild-type and *psf1-1* cells presented in Fig 6E.**
Contingency table and chi-square test.
(PDF)

**S3 Table. Yeast strains used in this work.**
(PDF)

**S4 Table. Primers used in this work.**
(PDF)

**S5 Table. Random sequences used in control experiments.**
(PDF)

## Acknowledgments

We thank Hiroyuki Araki (National Institute of Genetics, Mishima, Japan), Robert S. Lahue (National University of Ireland, Galway, Ireland), and Thomas D. Petes (Duke University School of Medicine, Durham, NC, USA) for providing strains and plasmids. We are grateful to our colleagues from the Laboratory of Mutagenesis and DNA Repair for fruitful discussions. We thank Malgorzata Lichocka from the Laboratory of Confocal and Fluorescence Microscopy for assistance with fluorescence microscopy and Jaroslaw Poznanski for help with statistical analyses.

## Author Contributions

**Conceptualization:** Malgorzata Alabrudzinska, Michal Dmowski, Iwona J. Fijalkowska.

**Data curation:** Malgorzata Jedrychowska, Milena Denkiewicz-Kruk, Malgorzata Alabrud-zinska, Adrianna Skoneczna, Michal Dmowski.

**Formal analysis:** Michal Dmowski, Iwona J. Fijalkowska.

**Funding acquisition:** Michal Dmowski, Iwona J. Fijalkowska.

**Investigation:** Malgorzata Jedrychowska, Milena Denkiewicz-Kruk, Malgorzata Alabrud-zinska, Adrianna Skoneczna, Piotr Jonczyk, Michal Dmowski.

**Supervision:** Michal Dmowski, Iwona J. Fijalkowska.

**Visualization:** Michal Dmowski.

**Writing – original draft:** Michal Dmowski, Iwona J. Fijalkowska.

**Writing – review & editing:** Michal Dmowski, Iwona J. Fijalkowska.

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
