## [Decision Letter · Decision Letter 0]

6 Sep 2019

Dear Dr Dmowski,

Thank you very much for submitting your Research Article entitled 'Defects in the GINS complex increase the instability of repetitive sequences via a recombination-dependent mechanism' to PLOS Genetics. Your manuscript was fully evaluated at the editorial level and by independent peer reviewers.

The reviewers consider the study to have potential, as do we. Reviewer raises a number of points that need correcting, while reviewer 2 has some more substantial criticisms, which we would like you address, if necessary experiementally, or explain why the points are invalid.

If you decide to revise the manuscript for further consideration at PLOS Genetics, please aim to resubmit within the next 60 days, unless it will take extra time to address the concerns of the reviewers, in which case we would appreciate an expected resubmission date by email to plosgenetics@plos.org.

[LINK]

We look forward to receiving your revised paper.

Yours sincerely,

Julian E. Sale

Associate Editor

PLOS Genetics

Gregory P. Copenhaver

Editor-in-Chief

PLOS Genetics

Reviewer's Responses to Questions

**Comments to the Authors:**

Reviewer #1: This manuscript describes the impact of psf1-1 mutation affecting a subunit of GINS complex on the stability of several types of repeated DNA sequences, including mono-, di-, trinucleotide repeats and minisatellites. The authors observe that the repeat instability is increased in the psf1-1 mutants and then proceed to combine the psf1-1 mutation with other DNA maintenance defects to address the mechanisms of instability. They find that mismatch repair prevents instability of microsatellites while proteins involved in homologous recombination (Rad51, Rad52), particularly BIR (Pol32, Pif1) and template switching (Mms2) promote instability. In addition, the authors detect the accumulation of single-stranded DNA and slightly increased levels of Rad51 in the psf1-1 cells. A model is proposed in which the replication disturbances resulting from GINS deficiency lead to the activation of HR pathway to rescue the collapsed forks, but errors during the homology search in the repetitive sequences or polymerase slippage during recombination-associated synthesis leads to expansion or contraction of the repeats. The work is technically excellent and well described. The novelty of findings is moderate as the effects of mismatch repair, HR, BIR and the other pathways on the stability of repeated DNA are generally known. The work does provide novel insights, however, in that it shows that the problems associated with replication of repetitive DNA are exacerbated in the psf1-1 mutants. This is important since GINS deficiency has recently been implicated in a hereditary disorder.

I suggest the following minor revisions to improve clarity:

1. Abstract can be shortened to reduce repetitions. For example, it is repeated three times that GINS is a component of CMG. Other suggestions for the abstract:

a) Line 32. It would be more logical to introduce the Saccharomyces cerevisiae model earlier, as the previous sentence also mentions the yeast Psf1-1. In the same sentence, “in psf1-1 mutants” seems to be missing.

b) The next sentence. I suggest changing “using psf1-1 derivatives lacking genes…” to “using derivatives of psf1-1 strains lacking genes…”

c) Lines 43-45. Although the idea is more or less clear, the sentence could be revised to reduce redundancy (repeat instability = expansion or contraction).

d) The last sentence. It is not clear whether the authors refer to recombination-associated synthesis or this is a separate mechanism of instability. The idea itself is also not novel, so emphasizing how this relates to the psf1-1 situation would make the sentence more relevant to the present work.

2. Author summary:

a) Lines 60-61. “repeated DNA sequences in mononucleotide, dinucleotide, trinucleotide and longer tracts” is a tautology.

b) Line 61. “Our results indicate…” could be changed to “Our results suggest…” as evidence provided is indirect.

c) Line 64. I suggest changing “synthesis via repetitive regions” to “synthesis through repetitive regions”.

3. Lines 82. Change “dinucleotides” to “dinucleotide repeats”. The same in line 84 for “mononucleotides”

4. Lines 106-112. This segment seems out of place. It is also confusing. The references provided to support the statement about “the first models proposed” are newer than the references that support the role of recombination, repair and transcription on lines 98-99.

5. On p. 6, there is no clear distinction between DNA maintenance systems that contribute to and that prevent instability. For example, if a DNA polymerase mutation increases instability (line 108), it is not appropriate to say that replication is normally involved in instability (lines 97-98).

6. Lines 124-126 and 173-175. I could not find the examples of various diseases promoted by CMG-E defects in the references provided.

7. Lines 139-141 and 302. The word “interestingly” can be deleted. Increased Pol zeta-dependent mutagenesis in replication mutants is a well-established phenomenon. I suggest changing the sentence to “As observed with many other replisome defects, a significant fraction of spontaneous mutagenesis in this mutant is due to…” Similarly, the effect of MMR on microsatellite instability is well-known.

8. Line 146. Change “participate in” to “contribute to”.

9. Lines 148-151. The sentence is confusing.

10. Figure legends should indicate what asterisks mean (what level of significance).

11. Subsection title “The psf1-1 allele causes increased instability of DNA repeat tracts” overlaps with the previous title “The psf1-1 allele enhances trinucleotide repeat expansion in yeast”, as TNRs are also repeats. The title(s) should be revised to remove ambiguity. The same comment applies to the titles of Figures 1 and 2.

12. Line 217. I would change “repetitive nature” to “repeats”.

13. Line 219. The explanation “which is toxic to cells producing the Ura3 protein [71]” would be more appropriate in the first section where the TNR assay is described as it uses the same selection.

14. The title of Figure 2. Delete the word “significantly”. It is assumed that increases that are not significant are not reported as increases.

15. Line 253. Delete “such as”.

16. Lines 257-259.

a) Move this segment one sentence up, as “these types” refers to alterations detected by electrophoresis, not to the mononucleotide run changes that could not be analyzed.

b) It is unclear what it means that the alterations are consistent with the structures. A more thorough explanation is needed.

c) Figure S1 has many details that are not explained in the legend. The letters designating DNA bases are also hard to read.

17. The subsection title “Unstable microsatellite sequences in psf1-1 cells are repaired by MMR:” may need to be revised. MMR repairs mismatches or loops but not sequences.

18. The title of Figure 3. “Impact of MSH2 disruption” could be deleted as it is redundant with “Effect of mismatch repair”. The same comment applies to Figures 4 and 7.

19. Line 365. Delete “as observed in”.

20. Line 376. I would change “deletion” to “loss” as “deletion” usually applies to genes/DNA and “player” must be a protein.

21. Increases in mutagenesis and synergistic or additive or epistatic interactions of mutator effects are seen with the control non-repetitive sequences in many experiments. These results are stated but not explained or discussed. It would be good if the authors could comment on the implications of these observations.

22. Line 406. I would change “This result demonstrates the accumulation of ssDNA stretches…” to “This result is consistent with the idea that ssDNA stretches accumulate…” As S1 nuclease also cuts at nicks, the results do not directly demonstrate the increase in ssDNA.

23. Line 436. I would revise “a mutator effect on forward URA3 mutagenesis”. In the same sentence, if the effect is not significant, it should be states as no increase rather than 1.6-fold increase.

24. Line 454. “Another mutagenic subpathway of HR” – in addition to what? Template switching addressed in the previous section is generally not mutagenic, at least outside the context of microsatellites. In the same sentence, it is unclear what “extensive synthesis of repetitive DNA sequences” is.

25. Lines 458-459. Pol32 is a subunit of Pol delta and is involved in all Pol delta-dependent processes. It does not seem appropriate to discuss it as BIR-specific protein. If there are BIR-specific effects of pol32 mutation, this should be explained in a less misleading manner.

26. Line 463. Change “allele… does not grow” to “strains… do not grow”.

27. Line 466. Was there a variety of phenotypes of the double mutants? What proportion were sensitive to both low and high temperature?

28. Sentences on lines 467 and 477 seem to contradict each other.

29. Line 531. “URA-“ mutants is not the proper yeast nomenclature.

30. Line 554. “Abolishes by 40-50%” may need to be revised, as does “mutator phenotype in the CAN1 reporter gene”.

31. Lines 562-563. To my knowledge, the study cited did not analyze duplications.

32. Line 674. The meaning of the phrase “replication is not entirely continuous” is not clear. Obviously, there are discontinuities between replicons and between Okazaki fragments. Do the authors refer to a single replicon and the leading strand? Even if so, the accumulation of ssDNA does not necessarily suggest discontinuity, it could be uncoupling of polymerase and helicase but ultimately continuous synthesis.

33. Lines 713 and 800. Change “nucleotides” to “nitrogenous bases”.

34. Line 736. All the strains listed are of “a” mating type, so it is not clear how the crosses were performed.

35. Line 756. “Lys2” should read “LYS2”.

36. Line 796. “SC8020” should read “SC802”.

37. The Discussion needs to be significantly shortened. There is a lot of redundancy between Results and Discussion. Much of Discussion is just re-statement of the results.

Reviewer #2: Jedrychowska et al tested the hypothesis that the GINS complex is important for repeat stability in yeast. To do so, they used a temperature sensitive mutant, psf1-1, which was known to destabilize the GINS complex and lead to increased mutagenesis load. They further implicated various repair pathways, from mismatch repair to homologous recombination, by generating double mutants. They tested several different repeat compositions, including disease causing trinucleotide repeats using chromosomal assays and other micro and minisatellites using a plasmid-based assay.

I have several reservations about the paper. First, it is not surprising that factors that drive mutagenesis would also drive instability of tandem repeats. Indeed, most of the findings are not specific to repetitive sequences. Second, the conclusions, including the title of the manuscript and the final models, fail to capture the complexity of the findings. Third, it is not clear to me whether the different pathways (MMR and BIR for example), work together. Finally, there are problems with statistical analyses and some of the methods used are not explained well enough to be able to determine whether they are valid.

Specific issues:

The plasmid based assay – the reason for using a plasmid based assay was because “Additionally, the disadvantage of the chromosomal trinucleotide repeat assay is the extremely high observed instability (two orders of magnitude higher than that in wild-type cells) of repeat tracts in psf1-1, making all manipulation, including the construction of double mutants, very risky.” This is not valid. Multiple mutants have been constructed in this assay, see for example Daee et al MCB 2007. Moreover, the plasmid assay was not used afterwards for trinucleotide repeats. Another issue that I have is that it is not clear whether the FOA resistant cells could arise simply from plasmid loss. A description of the plasmid system seems necessary – what is the origin of replication? Where within the URA3 sequence are the micro and minisatellites inserted? Are the authors detecting any repeat-induced mutations in addition to changes in repeat size? Or do they detect only changes in repeat tract size? Can the plasmid borne ura3 copy can recombine with the genomic one? This is not well explained. Some of this is in the paper they reference, but it seems to me that it would be better to have the information all in the same place.

The conclusions of the paper, that the GINS complex is important for repeat stability seems true. But the authors go further and add that this is dependent on homologous recombination. This is also true, but it is also true that it depends on MMS2, Pif1, and Pol32, and for some of them, Msh2. It is not clear how these other three requirements are genetically linked to Rad52 and Rad51 because there is no double or triple mutants made. Furthermore, none of these mutants were tested for trinucleotide repeats.

I would like to seem multiple mutant analysis so that we can have a better idea of what the mechanisms for instability are.

Statistics: The authors perform fluctuation analyses to determine the rate of mutation. This is the best way to go. But then they use a Mann-Whitney U-test to determine whether the changes are significant. It is not explained why they used this statistical analysis or what the input would be. I imagine that they needed to use the frequencies from the individual cultures, rather than the rate, for this analysis, which then would be comparing apples and oranges. The authors should report the 95% confidence intervals instead. Also the P-values are not reported, only stars were given.

Fig. 6: Statistics not explained in B and not done at all in E.

The authors claim that the mechanism is replication-dependent, but it was not formally shown. For instance, one would expect that keeping the cultures in stationary phase would not increase the frequency of instability if replication is indeed required.

Minor comments:

The Msh2 and Msh6 complex is called MutSalpha, the Msh2 / Msh3 one MutSbeta

I am tempted to argue that the paper would be more focused and easier for the reader to digest if it did not include the trinucleotide repeats. There is only data in the psf1-1 mutant.

Can the authors comment on why they did not use the non-permissive temperature for the psf1-1 mutant?

Add the random sequence used in the control plasmid in the supplementary.

**Have all data underlying the figures and results presented in the manuscript been provided?**

Reviewer #1: Yes

Reviewer #2: Yes

PLOS authors have the option to publish the peer review history of their article (what does this mean?). If published, this will include your full peer review and any attached files.

Reviewer #1: No

Reviewer #2: No

---

## [Editor Report · Decision Letter 1]

25 Oct 2019

Dear Dr Dmowski,

Thank you for submitting your revised manuscript and response to the reviewers' comments, which we have read. We are content that you have satisfactorily addressed the reviewers' comments and therefore are pleased to inform you that your manuscript "Defects in the GINS complex increase the instability of repetitive sequences via a recombination-dependent mechanism" has been editorially accepted for publication in PLOS Genetics. Congratulations!

Yours sincerely,

Julian E. Sale

Associate Editor

PLOS Genetics

Gregory P. Copenhaver

Editor-in-Chief

PLOS Genetics

Comments from the reviewers (if applicable):

**Data Deposition**

http://datadryad.org/submit?journalID=pgenetics&manu=PGENETICS-D-19-01301R1

Press Queries

---

## [Editor Report · Acceptance letter]

2 Dec 2019

PGENETICS-D-19-01301R1 

Defects in the GINS complex increase the instability of repetitive sequences via a recombination-dependent mechanism 

Dear Dr Dmowski, 

We are pleased to inform you that your manuscript entitled "Defects in the GINS complex increase the instability of repetitive sequences via a recombination-dependent mechanism" has been formally accepted for publication in PLOS Genetics! Your manuscript is now with our production department and you will be notified of the publication date in due course.

With kind regards,

Kaitlin Butler

PLOS Genetics

On behalf of:
